# Constrained Binary Decision Making

**Daniel Průša**    **Vojtěch Franc**
Department of Cybernetics
Faculty of Electrical Engineering
Czech Technical University in Prague
`{prusapa1,xfrancv}@fel.cvut.cz`

## Abstract

Binary statistical decision making involves choosing between two states based on statistical evidence. The optimal decision strategy is typically formulated through a constrained optimization problem, where both the objective and constraints are expressed as integrals involving two Lebesgue measurable functions, one of which represents the strategy being optimized. In this work, we present a comprehensive formulation of the binary decision making problem and provide a detailed characterization of the optimal solution. Our framework encompasses a wide range of well-known and recently proposed decision making problems as specific cases. We demonstrate how our generic approach can be used to derive the optimal decision strategies for these diverse instances. Our results offer a robust mathematical tool that simplifies the process of solving both existing and novel formulations of binary decision making problems which are in the core of many Machine Learning algorithms.

## 1   Introduction

Binary statistical decision making (BDM) involves selecting between two possible states using statistical evidence. This approach is applied across various fields, from classical statistics to machine learning. A notable example is the Neyman-Pearson problem [18], which establishes the optimal strategy for testing two hypotheses and serves as the foundation for optimal detectors. Beyond classical statistics, BDM problems are used in many machine learning applications, such as detecting out-of-distribution (OOD) samples [11, 14, 6, 5, 15, 10, 2, 21, 20, 22, 23], designing selective classifiers [13, 12, 4, 8, 9], and developing selective classification in the presence of OOD data (SCOD) [24, 16, 7] to name a few.

Instances of BDM determine the best decision-making strategy by solving a constrained optimization problem involving Lebesgue measurable functions in both the objective and constraints. Understanding the structure of this optimal strategy is crucial before deploying efficient methods to find it. The Neyman-Pearson lemma, a well-known example found in many statistics textbooks, states that an optimal detection strategy relies on comparing the likelihood ratio of the underlying distributions with a threshold. Therefore, learning a detector from examples involves effectively approximating the likelihood ratio and then tuning the decision threshold.

However, defining the optimal strategy for various BDM problems is generally challenging, often taking several years to discover. For instance, the Bounded-Improvement and Bounded-Abstention models, serving as formal definitions of optimal selective classifiers, were first introduced in [19]. The optimal strategies for these two BDM problems were initially described in [8], marking a 14-year period of development. Another recent example is the work of [24], who developed a method for learning a selective classifier handling inputs contaminated by out-of-distribution data, so called SCOD problem. Although they formulated an optimal SCOD selection strategy to accept input samples as a solution to a novel BDM problem, they did not analyze the optimal strategy directly.

38th Conference on Neural Information Processing Systems (NeurIPS 2024).

Instead, they proposed a selector strategy named SIRC, combining two detectors using heuristically chosen non-linear functions. In a recently published paper [16], the authors derived an optimal solution for the SCOD problem, demonstrating that the optimal selective strategy involves a simple linear combination of the two detectors. Empirical evidence confirmed that the optimal linear strategy consistently outperforms the heuristic SIRC, despite its minor change. This example underscores the advantages of understanding the structure of the optimal solution to underlying BDM problems.

In this paper, we present a comprehensive formulation of the BDM problem and thoroughly characterize the optimal strategy. Our framework encompasses various BDM problems as special cases, enabling us to derive optimal decision strategies for these instances. This provides a robust mathematical tool for solving both existing and new BDM problems. The related theorem is highly general, applying to both discrete and continuous instance spaces without requiring the differentiability of decision and loss functions, unlike common proof techniques based on Lagrange duality.

To illustrate the impact of our work, in Section 2, we formulate several practical modifications of existing BDM problems. In Section 4, we demonstrate the derivation of their optimal strategies using our main result, which characterizes the optimal strategies of a generic BDM as presented in Section 3.

## 2 Examples of existing binary decision making problems

In this section, we present various examples of BDM problems, with a focus on the significant application of designing a selective classifier in the presence of out-of-distribution (SCOD) samples [24, 16]. We note that while selective classification itself is not the contribution of this paper, it serves to illustrate the practical importance and diversity of BDM formulations in addressing a recently emerging problem in the machine learning community.

Consider a given classifier $h\colon \mathcal{X} \to \mathcal{Y}$, where $\mathcal{X}$ is the instance space and $\mathcal{Y}$ a finite set of labels. We assume the classifier $h$ was designed to minimize the prediction loss $\ell\colon \mathcal{Y} \times \mathcal{Y} \to \mathbb{R}$. Our task is to equip the classifier $h$ with a binary stochastic decision strategy $c\colon \mathcal{X} \to [0, 1]$. The strategy $c$ acts as a selector for input samples. The input sample $x \in \mathcal{X}$ is accepted for prediction by $h$ with probability $c(x)$ and is rejected otherwise. Thus, the pair $(h, c)\colon \mathcal{X} \to \mathcal{Y} \cup \{\text{reject}\}$ forms a reject option classifier [3]:

$$(h, c)(x) = \left\{ \begin{array}{ll} h(x) & \text{with probability} \quad c(x)\,, \\ \text{reject} & \text{with probability} \quad 1 - c(x)\,. \end{array} \right.$$

We assume the following statistical model of the data. Input samples are generated from a mixture of two distributions: the in-distribution (ID) $p_I\colon \mathcal{X} \to \mathbb{R}_+$ and the out-of-distribution (OOD) $p_O\colon \mathcal{X} \to \mathbb{R}_+$. Thus, the input sample $x \in \mathcal{X}$ is generated from

$$p(x) = p_I(x)\,(1 - \pi) + p_O(x)\,\pi\,, \tag{1}$$

where $\pi \in [0, 1]$ represents the portion of OOD samples in the mixture. Furthermore, if the sample $x \in \mathcal{Y}$ is generated from the in-distribution, a latent label $y \in \mathcal{Y}$ is generated according to the distribution $p(y \mid x)$.

In the outlined statistical model, prediction uncertainty arises from two main sources: i) inherent data uncertainty (aleatoric uncertainty), stemming from the stochastic relationship between the input $x$ and the latent label $y$, and ii) distribution uncertainty, which arises from input samples potentially coming from either in-distribution or out-of-distribution sources. We will explore six different BDM instances to develop a selector $c$ that admits an input sample $x$ for classification with $h$ only when the prediction uncertainty is below an acceptable threshold. Each instance defines an optimal selector in a unique way. In Section 2.1, we described the Neyman-Pearson formulation of the OOD optimal detector [18], which addresses only distribution uncertainty. Sections 2.2 and 2.3 cover the Bounded-Improvement and Bounded-Abstention models [19], addressing only data uncertainty. In Section 2.4, we describe the SCOD problem [24], which defines the optimal selector addressing both distribution and data uncertainty. Additionally, Sections 2.5 and 2.6 give examples of two novel practical variants of the SCOD problem.

The primary contribution of this paper is a framework for solving a broad range of BDM problems. In Section 3, we detail the main theorem that characterizes the optimal solution for a generic BDM problem. In Section 4, we demonstrate how to apply this theorem to find solutions easily for all the BDM instances described here.

## 2.1 Neyman-Pearson (NP) detector

Assume our goal is to design a selector strategy $c\colon \mathcal{X} \to [0,1]$ that accepts only in-distribution (ID) samples and rejects out-of-distribution (OOD) samples. In other words, $c$ functions as a binary classifier to distinguish ID and OOD samples. We will characterize the strategy $c$ using two metrics. First, using the True Positive Rate (TPR) representing the probability that ID sample $x$ is correctly accepted by the strategy:

$$\text{tpr}(c) = \int_{\mathcal{X}} p_I(x)\, c(x)\, dx\,. \tag{2}$$

Second, using the False Positive Rate (FPR) representing the probability that OOD sample $x$ is incorrectly accepted by $c$:

$$\text{fpr}(c) = \int_{\mathcal{X}} p_O(x)\, c(x)\, dx\,. \tag{3}$$

Given maximal acceptable FPR, $\text{fpr}_{\max} > 0$, the NP task is to find a strategy $c^*$ that solves the following BDM problem [18]:

$$\max_{c\colon X \to [0,1]} \text{tpr}(c) \quad \text{s.t.} \quad \text{fpr}(c) \le \text{fpr}_{\max}\,. \tag{4}$$

A well-known solution to the NP task 4 is the strategy:

$$c^*(x) = \begin{cases} 1 & \text{if} \quad g(x) < \theta\,, \\ \tau & \text{if} \quad g(x) = \theta\,, \\ 0 & \text{if} \quad g(x) > \theta\,, \end{cases} \tag{5}$$

where $g(x) = \frac{p_O(x)}{p_I(x)}$ is the likelihood ratio, $\theta \in \mathbb{R}$ is a decision threshold, and $\tau \in [0,1]$ is a probability of acceptance when $g(x) = \theta$.

Knowing the structure of the optimal strategy simplifies its construction significantly. The key to an optimal decision is accurately modeling the likelihood ratio $g(x)$. Once the likelihood ratio is known, finding the optimal strategy involves setting the threshold $\theta$ and the randomization parameter $\tau$, typically done by tuning these parameters on a validation sample. Moreover, if $p_I(x)$ and $p_O(x)$ are continuous, the randomization parameter $\tau$ can be arbitrary since the event $g(x) = \theta$ has zero probability of occurring.

## 2.2 Selective Classification: Bounded-abstention model

Let us consider the ideal scenario where the input instances $x \in \mathcal{X}$ are solely generated by the in-distribution $p_I(x)$, meaning the out-of-distribution portion in the model (1) is $\pi = 0$. Our aim is devise a selection strategy $c\colon \mathcal{X} \to [0,1]$ that accepts only those input samples $x \in \mathcal{X}$ where the prediction loss $\ell(y, h(x))$ is likely to be small. We will characterize the strategy $c$ using two metrics. First, we employ the notion of *coverage*:

$$\phi(c) = \int_{\mathcal{X}} p_I(x)\, c(x)\, dx \tag{6}$$

which reflects the probability of accepting an input sample. Secondly, we introduce the concept of "selective risk":

$$\text{R}^{\text{S}}(h, c) = \frac{\int_{\mathcal{X}} \sum_{y \in \mathcal{Y}} p_I(x, y)\, \ell(y, h(x))\, c(x)\, dx}{\phi(c)}\,. \tag{7}$$

representing the expected prediction loss $\ell\colon \mathcal{Y} \times \mathcal{Y} \to \mathbb{R}$ on the accepted samples.

Given a minimal acceptable coverage $\omega > 0$, the bounded-abstention model defines the optimal selector strategy as the solution to the following BDM problem [19]:

$$\min_{h \in \mathcal{Y}^{\mathcal{X}}, c \in [0,1]^{\mathcal{X}}} \text{R}^{\text{S}}(h, c) \quad \text{s.t.} \quad \phi(c) \ge \omega\,. \tag{8}$$

The solution to the bounded-abstention model (8) consists of the Bayes predictor

$$h^*(x) = \text{argmin}_{y'} \sum_y p_I(y \mid x) \ell(y, y')$$

and the strategy [8, 9]:

$$c^*(x) = \begin{cases} 1 & \text{if} \quad r(x) < \theta \,, \\ \tau & \text{if} \quad r(x) = \theta \,, \\ 0 & \text{if} \quad r(x) > \theta \,, \end{cases} \tag{9}$$

where

$$r(x) = \sum_{y \in \mathcal{Y}} p_I(y \mid x)\, \ell(y, h(x)) \tag{10}$$

is the conditional risk of the classifier $h$ for the input $x$, $\theta \in \mathbb{R}$ is a decision threshold, and $\tau \in [0, 1]$ is a probability of acceptance when $r(x) = 0$.

Note that the BDM problem (8) was formulated in [19], but the optimal solution (9) was not derived until 14 years later in [8].

The values of $\theta$ and $\tau$ depend on $p_I(x, y)$ and $\omega$, and in practice, they are tuned on validation data once a good model of the conditional risk $r(x)$ is established. While this process can be challenging and needs be addressed for each problem separately, it is still significantly easier than tuning an unknown function. For example, discretizing the possible values of $\theta$ and $\tau$ and conducting an exhaustive search, although not always optimal, has proven effective in practice.

The additivity of the risk $R^S$ allows $h$ to be optimized independently of $c$, resulting in the Bayes predictor for all problems discussed in the following sections. Therefore, from this point on, we will focus solely on optimal strategies for $c$.

## 2.3 Selective Classification: Bounded-improvement rejection model

A symmetric definition of the optimal selector strategy is provided by the bounded-improvement model. Given a maximum acceptable selective risk $\lambda > 0$, the optimal selector is the solution to the following BDM problem:

$$\max_{h,c} \phi(c) \quad \text{s.t.} \quad R^S(h, c) \le \lambda \,. \tag{11}$$

The solution to the bounded-improvement model (11) is similar to the strategy (9), differing only in the specific values of the decision threshold $\theta$ and the randomization parameter $\tau$ [9].

## 2.4 Selective Classification in the presence of OOD data (SCOD)

The SCOD problem integrates the objectives of the Neyman-Pearson task and bounded-abstention models. The aim is to design a selector strategy $c \in \mathcal{X} \to [0, 1]$ that accepts an input sample $x$ if it is likely to be correctly predicted by $h$ and unlikely to be generated from the OOD. Formally, the objective is defined using three metrics: True Positive Rate $\text{tpr}(c)$, equation (2), False Positive Rate $\text{fpr}(c)$, equation (3) and selective risk $R^S(h, c)$, equation (7). Given the relative cost $\alpha > 0$ and the minimum acceptable TPR, $\text{tpr}_{\min} > 0$, the goal is to find a strategy $c^*$ that solves the following BDM problem [24]:

$$\min_{h \in \mathcal{Y}^{\mathcal{X}}, c \in [0,1]^{\mathcal{X}}} \left[ (1 - \alpha)\, R^S(h, c) + \alpha\, \text{fpr}(c) \right] \quad \text{s.t.} \quad \text{tpr}(c) \ge \text{tpr}_{\min} \,. \tag{12}$$

The solution to the SCOD problem (12) is the strategy [16, 7]:

$$c^*(x) = \begin{cases} 1 & \text{if} \quad s(x) > \theta \,, \\ \tau & \text{if} \quad s(x) = \theta \,, \\ 0 & \text{if} \quad s(x) < \theta \end{cases} \quad \text{where} \quad s(x) = r(x) + \frac{\alpha\, \text{tpr}_{\min}}{1 - \alpha} g(x) \tag{13}$$

for $\alpha \in [0, 1)$ and $s(x) = g(x)$ for $\alpha = 1$. Here, $r(x)$ is the conditional risk (10), $g(x) = \frac{p_O(x)}{p_I(x)}$ is the likelihood ratio, $\theta \in \mathbb{R}$ is a decision threshold, and $\tau \in [0, 1]$ the acceptance probability for the case $s(x) = r(x)$.

The BDM problem (12) was formulated in [24]. The optimal solution for continuous distributions was recently derived in [16], and the solution for the general case was provided in [7].

## 2.5 SCOD: bounded TPR-FPR

To demonstrate the versatility of our framework, we will present two novel variants of the SCOD problem in in this section and the next. While the optimal solutions to these new formulations are unknown, they can be easily derived using the framework proposed in this paper, as we will later show.

The original formulation of the SCOD problem (12) relies on the user-defined relative cost $\alpha$. However, this assumes that the prediction loss $\ell \colon \mathcal{Y} \times \mathcal{Y} \to \mathbb{R}$ and the cost for incorrectly accepting the OOD sample share the same units, which is not always practical. In such cases, it can be useful to remove the FPR from the objective and impose a hard constraint on it instead. Specifically, we can alternatively formulate the SCOD problem as follows. Given $\text{tpr}_{\min} > 0$ and $\text{fpr}_{\max} > 0$, the goal is to find a strategy $c^*$ that solves the following BDM problem:

$$\min_{h \in \mathcal{Y}^{\mathcal{X}}, c \in [0,1]^{\mathcal{X}}} \mathrm{R}^{\mathrm{S}}(h, c) \qquad \text{s.t.} \qquad \text{tpr}(c) \geq \text{tpr}_{\min} \quad \text{and} \quad \text{fpr}(c) \leq \text{fpr}_{\max} . \qquad (14)$$

In Section 4, we will prove that the optimal solution to the problem (14) is the strategy:

$$c^*(x) = \begin{cases} 1 & \text{if} \quad s(x) < \theta \,, \\ \tau & \text{if} \quad s(x) = \theta \,, \\ 0 & \text{if} \quad s(x) > \theta \,, \end{cases} \qquad \text{where} \quad s(x) = r(x) + \beta g(x) \,, \qquad (15)$$

with $\theta \in \mathbb{R}$ as the decision threshold, $\tau \in [0, 1]$ as the acceptance probability when $s(x) = \theta$, and $\beta \in \mathbb{R}$ as a constant depending on the problem setup.

## 2.6 SCOD: bounded Precision-Recall

Assume the portion of OOD samples $\pi$ is known or can be estimated. In some applications, it is useful to replace FPR with precision, which is the proportion of true ID samples among all accepted inputs. Precision $\text{prec}(c)$ is defined as:

$$\text{prec}(c) = \frac{(1 - \pi) \, \text{tpr}(c)}{\text{fpr}(c) \, \pi + \text{tpr}(c) \, (1 - \pi)} \,.$$

If we want to avoid defining the relative cost, as in (14), we can formulate the SCOD problem as follows. Given $\text{tpr}_{\min} > 0$ and $\text{prec}_{\min} > 0$, the goal is to find a strategy $c^*$ which solves the following BDM problem:

$$\min_{h \in \mathcal{Y}^{\mathcal{X}}, c \in [0,1]^{\mathcal{X}}} \mathrm{R}^{\mathrm{S}}(h, c) \quad \text{s.t.} \qquad \text{tpr}(c) \geq \text{tpr}_{\min} \quad \text{and} \quad \text{prec}(c) \geq \text{prec}_{\min} \qquad (16)$$

In Section 4, we will prove that the optimal solution to the problem (16) is a strategy similar to (15), differing only in the specific values of the decision threshold $\tau$, the randomization parameter $\tau$ and the multiplier $\beta$.

## 2.7 Summary

In the previous sections, we described six BDM problems, each with a unique optimal selector strategy. The first four are established formulations, while the last two are novel SCOD modifications potentially useful for specific applications. As ML applications grow, new formulations are likely to emerge to address specific setups.

Deriving the optimal strategy for a given BDM problem proved to be the essential first step. Although the statistical model is often unknown in practice, understanding the structure of the optimal strategy is crucial for designing efficient algorithms to learn the selector from examples. This knowledge can narrow the search space or simplify the problem using divide-and-conquer approaches.

For instance, consider constructing a selector strategy for the SCOD problem. Whether using the original cost-based formulation (12) or the two novel variants (14) and (16), takes the form (15), the optimal strategy $c^*$ relies on an uncertainty score $s(x) = r(x) + \beta g(x)$. The strategy accepts input $x$ if $s(x)$ is below a threshold $\theta$, and with probability $\tau$ if $s(x) = \theta$. This analysis shows that the core problem reduces to approximating the two components: the conditional risk

$$r(x) = \sum_{y \in \mathcal{Y}} p(y \mid x) \ell(y, h(x))$$

and the OOD/ID likelihood ratio $g(x) = p_O(x)/p_I(x)$. Extensive literature on OOD detection provides methods to approximate $g$ (e.g. [11, 14, 6, 5, 15, 10, 2, 21, 20, 22, 23]), and there are methods to construct good proxies for $r$ (e.g. [13, 12, 4, 8, 9]).

Once $r$ and $g$ are known, constructing the selector strategy involves determining the scalars $\theta$, $\tau$, and $\beta$. These parameters can be tuned using held-out data, which is much easier than solving the original problem. Moreover, for continuous distributions, the event $s(x) = \theta$ has zero probability, making $\tau$ arbitrary. In the cost-based formulation, the multiplier $\beta$ has an analytical solution, leaving only $\theta$ to be determined.

This example shows that deriving an optimal strategy for BDM problems allows us to use existing methods from OOD detection and selective classification to approximate various SCOD problem formulations. Conversely, skipping the optimal strategy derivation and using heuristic rules, such as the SIRC strategy from the original SCOD paper [24], can lead to sub-optimal performance [16].

## 3    Main result

In this section, we formulate a generic BDM problem, characterize its optimal solutions, and provide a straightforward instance of the optimal strategy.

For a measurable set $\mathcal{X}$ in a measure space, Lebesgue measurable functions $R, p, q : \mathcal{X} \to [0, \infty)$ with finite integrals on $\mathcal{X}$, and fixed non-negative real values $\delta$ and $\sigma$, we define the following optimization problem:

$$\min_{c:\ \mathcal{X} \to [0,1]} \int_{\mathcal{X}} R(x)c(x)dx \quad \text{s.t.} \quad \int_{\mathcal{X}} p(x)c(x)dx \geq \delta \quad \text{and} \quad \int_{\mathcal{X}} q(x)c(x)dx \leq \sigma \,. \tag{17}$$

Note that we do not pose any additional requirements on the functions, like continuity or differentiability, neither do we differentiate between discrete and continuous sets $\mathcal{X}$; they can be arbitrary.

**Theorem 1.** *For every optimal solution $c^* : \mathcal{X} \to [0, 1]$ to Problem* (17)*, there exist real numbers $\lambda, \mu$ such that*

$$\int_{\mathcal{X}^<} p(x)c^*(x)\, dx = \int_{\mathcal{X}^<} p(x)\, dx \,, \tag{18}$$

$$\int_{\mathcal{X}^>} p(x)c^*(x)\, dx = 0 \,, \tag{19}$$

*where*

$$\mathcal{X}^< = \left\{ x \in \mathcal{X} \mid p(x) > 0 \wedge \frac{R(x)}{p(x)} + \mu\frac{q(x)}{p(x)} < \lambda \right\} \,, \tag{20}$$

$$\mathcal{X}^> = \left\{ x \in \mathcal{X} \mid p(x) > 0 \wedge \frac{R(x)}{p(x)} + \mu\frac{q(x)}{p(x)} > \lambda \right\} \,. \tag{21}$$

Informally, equation (18) states that any optimal $c^*$ attains the value 1 on $\mathcal{X}^<$, except on a subset of measure zero. Similarly, by equation (19), $c^*$ attains the value 0 on $\mathcal{X}^>$, again up to a subset of measure zero. The sets $\mathcal{X}^<$ and $\mathcal{X}^>$ contain the points $x$ for which $\frac{R(x)}{p(x)} + \mu\frac{q(x)}{p(x)}$ is below and above the threshold $\lambda$, respectively, excluding any insignificant elements where $p(x) = 0$.

Based on these observations, the next lemma suggests a way to construct a single score function that determines an optimal decision strategy.

**Lemma 1.** *An optimal solution $c^* : \mathcal{X} \to [0, 1]$ to Problem* (17) *can be found among the forms*

$$c^*(x) = \begin{cases} 0 & \text{if } s(x) > \lambda \,, \\ \tau(x) & \text{if } s(x) = \lambda \,, \\ 1 & \text{if } s(x) < \lambda \,, \end{cases}$$

*where*

$$s(x) = \begin{cases} \frac{R(x)}{p(x)} + \mu\frac{q(x)}{p(x)} & \text{if } p(x) > 0 \,, \\ \infty & \text{otherwise,} \end{cases}$$

*is a score function, $\lambda, \mu$ are suitable real numbers, and $\tau : \mathcal{X} \to [0,1]$ is a function implicitly defined by the problem parameters.*

*Proof.* Let $c : \mathcal{X} \to [0,1]$ be an optimal solution to Problem (17). Consider real numbers $\lambda, \mu$ yielded by Theorem 1 and define

$$c^*(x) = \begin{cases} 0 & \text{if } s(x) > \lambda, \\ c(x) & \text{if } s(x) = \lambda, \\ 1 & \text{if } s(x) < \lambda. \end{cases}$$

To show that $c^*$ is optimal, decompose $\mathcal{X}$ into the pairwise disjoint sets $\mathcal{X}^<$ (see (20)), $\mathcal{X}^>$ (see (21)), $\mathcal{X}' = \{x \in \mathcal{X} \mid p(x) = 0\}$ and $\mathcal{X}^= = \{x \in \mathcal{X} \mid s(x) = \lambda\}$. Equalities (18) and (19) ensure that $c$ and $c^*$ are nearly identical on sets $\mathcal{X}^<$ and $\mathcal{X}^>$, respectively, up to subsets of measure zero. By the definition of $c^*$, they are identical on $\mathcal{X}^=$. Consequently, the criterion and constraints attain the same values for both $c$ and $c^*$ on $\mathcal{X}^< \cup \mathcal{X}^> \cup \mathcal{X}^=$. Finally, the first constraint of (17) is independent of $\mathcal{X}'$, and $c^*(x) \leq c(x)$ for all $x \in \mathcal{X}'$ ensures

$$\int_{\mathcal{X}'} R(x)c^*(x) \leq \int_{\mathcal{X}'} R(x)c(x), \quad \int_{\mathcal{X}'} q(x)c^*(x) \leq \int_{\mathcal{X}'} q(x)c(x).$$

In conclusion, $c^*$ is an optimal solution. $\qquad\square$

**Tool to solve the BDM problems**  Lemma 1 is a key tool for simplifying the BDM problem (17). It shows that the main challenge is approximating the ratios $\frac{R(x)}{p(x)}$ and $\frac{q(x)}{p(x)}$. Once these ratios are known, determining the score $s$ involves finding the multiplier $\mu$. With the score $s$, the optimal strategy $c^*$ depends on finding the decision threshold $\lambda$ and the function $\tau(x)$. However, in the cases where $\mathcal{X}$ is a continuous space, the boundary set $\{x \in \mathcal{X} \mid s(x) = \lambda\}$ typically has measure zero, except in some pathological instances. Consequently, $\tau(x)$ can be chosen arbitrarily and does not play a significant role. Thus, in continuous spaces, one only needs to find the scalars $\mu$ and $\lambda$, which can be tuned using held-out data, making the process much simpler than solving the original problem from scratch.

**The problem feasibility**  In the special case where only one constraint is given, i.e., when either $p(x) = 0$ or $q(x) = 0$, checking feasibility is straightforward. In the general case, to determine the feasibility of Problem (17), we can formulate the task as:

$$\min_{c : \mathcal{X} \to [0,1]} \int_{\mathcal{X}} q(x)c(x)dx \quad \text{s.t.} \quad \int_{\mathcal{X}} p(x)c(x)dx \geq \delta. \tag{22}$$

We obtain a similar problem but with only one constraint. Problem (17) is feasible if and only if Problem (22) attains a value less than or equal to $\sigma$.

**Proof technique**  Problem (17) represents an instance of infinite linear programming, grounded in established theory [17]. Especially, when $\mathcal{X}$ is a finite set, it reduces to a standard linear program, allowing the use of Lagrange duality to derive optimality conditions, as characterized in Theorem 1. However, when $\mathcal{X}$ is arbitrary and the functions $R, p, q$ are essentially unrestricted, this approach becomes substantially more complex. In such cases, a more general duality theorem would be required, and even then, deducing the desired results would involve handling significantly more intricate optimality conditions. Given these challenges and drawing inspiration from techniques in related works (e.g. [9, 16, 18] and other studies on the Neyman-Pearson problem), we instead pursue a direct proof, provided in Appendix A. By introducing forbidden point configurations, this proof shows that each optimal solution is determined by a line separating the images of the sets $\mathcal{X}^<$ and $\mathcal{X}^>$ in the plane, mapped by the function $\varkappa(x) = \left( \frac{R(x)}{p(x)}, \frac{q(x)}{p(x)} \right)$.

**Extensions**  It remains an open problem whether our results can be generalized to the extended formulation

$$\min_{c : \mathcal{X} \to [0,1]} \int_{\mathcal{X}} R(x)c(x), dx \quad \text{s.t.} \quad \int_{\mathcal{X}} p(x)c(x), dx \geq \delta \quad \text{and} \quad \int_{\mathcal{X}} q_i(x)c(x), dx \leq \sigma_i, \quad i \in I, \tag{23}$$

where $I$ is a finite index set, the functions $q_i$ are Lebesgue measurable with finite integrals on $\mathcal{X}$, and $\sigma_i$ are non-negative bounds. A key question is whether there is a scoring function with $1 + |I|$ parameters (instead of 2) for this case. It is important to note that generalizing the current proof would involve addressing a significantly greater combinatorial complexity in the point configurations.

# 4 General tasks

In this section, we introduce several special cases of problem (17), whose forms of solutions directly follow from Theorem 1, as we will show. These cases encompass all the examples in Section 2. Thus, all the mentioned BDM problems can be solved in their general form using the single framework proposed in this paper.

To simplify notation, we represent the Lebesgue measurable functions in (17) as $p_i \colon \mathcal{X} \to [0, \infty)$, $i \in \{0, 1, 2\}$. Furthermore, for Lebesgue measurable $c \colon \mathcal{X} \to [0, 1]$, we define the functionals:

$$F_i(c) = \int_{\mathcal{X}} p_i(x) \, c(x) \, dx \,, \quad i \in \{0, 1, 2\} \,.$$

The generic BDM problem (17) then requires solving for given non-negative scalars $\delta$ and $\rho$:

$$\min_{c \colon \mathcal{X} \to [0,1]} F_0(c) \quad \text{s.t.} \quad F_1(c) \geq \delta \quad \text{and} \quad F_2(c) \leq \sigma \,. \tag{24}$$

Note that $F_0$ here does not depend on the classier $h$, which is consistent with the assumption that $h$ is the Bayes predictor.

**Problem 1:** Given $\delta > 0$, the task is to solve:

$$\min_{c \colon \mathcal{X} \to [0,1]} F_0(c) \quad \text{s.t.} \quad F_1(c) \geq \delta \,. \tag{25}$$

A special case is the Neyman-Pearson task (4).

We derive this formulation from (24) by setting $p_2(x) = 0$ for all $x \in \mathcal{X}$. Lemma 1 implies that if the problem is feasible, it has an optimal solution $c^*$ such that

$$c^*(x) = \begin{cases} 0 & \text{if } \frac{p_0(x)}{p_1(x)} > \lambda \,, \\ \tau(x) & \text{if } \frac{p_0(x)}{p_1(x)} = \lambda \,, \\ 1 & \text{if } \frac{p_0(x)}{p_1(x)} < \lambda \,. \end{cases}$$

Note that here, Lemma 1 does not fully replicate the Neyman-Pearson strategy, in which $\tau$ is constant.

**Problem 2:** Given $\delta > 0$, the task is to solve:

$$\min_{c \colon \mathcal{X} \to [0,1]} \frac{F_0(c)}{F_1(c)} \quad \text{s.t.} \quad F_1(c) \geq \delta \,. \tag{26}$$

Special cases are the Bounded-Abstention model (8) and the original SCOD problem (12).

This formulation can equivalently be transformed to (25) by showing that there is an optimal solution $c^*$ such that $F_1(c^*) = \delta$. Indeed, if some optimal $c \colon \mathcal{X} \to [0, \infty)$ fulfills $F_1(c) > \delta$, then we can define $c^* = \frac{\delta}{F_1(c)} c$, and it holds that $F_1(c^*) = \delta$ and

$$\frac{F_0(c^*)}{F_1(c^*)} = \frac{\frac{\delta}{F_1(c)} F_0(c)}{\delta} = \frac{F_0(c)}{F_1(c)} \,.$$

Hence, such a $c^*$ is preserved in the set of optimal solutions of the problem

$$\min_{c \colon \mathcal{X} \to [0,1]} \frac{F_0(c)}{\delta} \quad \text{s.t.} \quad F_1(c) \geq \delta \,.$$

**Problem 3:** Given $\delta > 0$, the task is to solve:

$$\max_{c \colon \mathcal{X} \to [0,1]} F_1(c) \quad \text{s.t.} \quad \frac{F_0(c)}{F_1(c)} \leq \delta \,. \tag{27}$$

A special case is the Bounded-Improvement model (11).

Let $c^*$ be an optimal solution to (27). Define a problem:

$$\min_{c:\ \mathcal{X}\to[0,1]} \frac{F_0(c)}{F_1(c)} \qquad \text{s.t.} \qquad F_1(c) \geq F_1(c^*)\,.$$

Clearly, $c^*$ is feasible for the new problem, which is of the form (26). Moreover, any optimal solution $c$ to this problem is also optimal to (27) because it attains the maximum $F_1(c^*)$ and satisfies the constraint:

$$\frac{F_0(c)}{F_1(c)} \leq \frac{F_0(c^*)}{F_1(c^*)} \leq \delta\,.$$

**Problem 4:**  Given $\delta > 0$, $\sigma > 0$, the task is to solve:

$$\min_{c:\ \mathcal{X}\to[0,1]} \frac{F_0(c)}{F_1(c)} \qquad \text{s.t.} \qquad F_1(c) \geq \delta \quad \text{and} \quad F_2(c) \leq \sigma\,. \tag{28}$$

A special case is the SCOD problem formulation with constraints on FPR and TPR (14).

In this case, we can transform the problem to (24) in the same way as we did it for the formulation (26).

**Problem 5:**  Given $\delta > 0$, $\sigma > 0$, the task is to solve:

$$\min_{c:\ \mathcal{X}\to[0,1]} \frac{F_0(c)}{F_1(c)} \qquad \text{s.t.} \qquad F_1(c) \geq \delta \quad \text{and} \quad \frac{F_2(c)}{F_1(c)} \leq \sigma\,. \tag{29}$$

A special case is the SCOD problem formulation with the constraints on TPR and Precision (16) because the second constraint in (16) can be rewritten as

$$\frac{\text{fpr}(c)}{\text{tpr}(c)} \leq (\pi^{-1} - 1)(\text{prec}_{\min}^{-1} - 1)\,,$$

which matches the form

$$\frac{F_2(c)}{F_1(c)} \leq \sigma\,.$$

Again, the problem transforms to (24) using the same approach as in the case of the formulation (26).

## 5   Conclusion

In this paper, we presented a comprehensive framework for solving binary decision-making (BDM) problems, which define optimal decision strategies through constrained optimization involving Lebesgue measurable functions. We characterize all optimal strategies for a generic BDM problem and derive a specific class of optimal strategies based on comparing a single score with a decision threshold. Our framework covers a variety of BDM problems, from well-known instances like the Neyman-Pearson problem to recent developments such as the SCOD problem. We demonstrated the versatility and robustness of our framework by deriving optimal solutions for all the BDM problems discussed in our paper. This work provides a foundational approach that can be adapted to future needs in machine learning and statistical decision making, ensuring more effective and theoretically grounded solutions.

## 6   Acknowledgement

This research was supported by the CTU institutional support (future fund).

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

# A   Appendix (proof of the main theorem)

For a metric vector space $V$ and its subset $A \subseteq V$, we write $\mathrm{conv}(A)$, $\mathrm{int}(A)$ and $\mathrm{span}(A)$ to denote the convex hull, interior, and span of $A$, respectively.

**Lemma 2.** *Let $A$ be a subset of $\mathbb{R}^2$ with $\dim \mathrm{span}(A) = 2$. Given a point $x$ located in the interior of the convex hull of $A$, there exists a convex polygon $P$ such that $x$ lies in the interior of $P$. Moreover, all vertices of the polygon $P$ are elements of the set $A$.*

*Proof.* Given that $x \in \mathrm{int}(\mathrm{conv}(A))$, there is a triangle $T \subseteq \mathrm{conv}(A)$ such that $x \in \mathrm{int}(T)$. According to Carathéodory's theorem [1], each vertex of $T$ lies in some simplex whose vertices are in the set $A$. Let these simplices be denoted as $S_1$, $S_2$, and $S_3$. Defining $P = \mathrm{conv}(S_1 \cup S_2 \cup S_3)$ forms a convex polygon $P$ with vertices in $A$, and $x$ is in the interior of $P$.  $\square$

*Proof of Theorem 1.* Let $\overline{\mathcal{X}} = \{x \in \mathcal{X} \mid p(x) > 0\}$. For all $x \in \overline{\mathcal{X}}$, we define $g(x) = \frac{q(x)}{p(x)}$ and $r(x) = \frac{R(x)}{p(x)}$. We will associate elements of $\overline{\mathcal{X}}$ with points in a plane by introducing the mapping $\varkappa : \overline{\mathcal{X}} \to \mathbb{R}_+^2$ such that $\varkappa(x) = (g(x), r(x))$.

The open $\varepsilon$-ball of $\pi(x)$ in $\mathbb{R}^2$ is denoted as $B_\varepsilon(x)$.

For a measurable $A \subseteq \mathbb{R}_+^2$, define

$$p(A) = \int_{\varkappa^{-1}(A)} p(x)dx \,, \tag{30}$$

$$c(A) = \begin{cases} \frac{1}{p(A)} \int_{\varkappa^{-1}(A)} p(x)c(x)dx & \text{if } p(A) > 0 \,, \\ 0 & \text{otherwise.} \end{cases} \tag{31}$$

Problem (17) depends only on the subset

$$\mathcal{X}^+ = \left\{ x \in \overline{\mathcal{X}} \mid \forall \varepsilon > 0 : p(B_\varepsilon(x)) > 0 \right\} \subseteq \mathcal{X}.$$

To see this, let us define $Y_1 = \{x \in \mathcal{X} \mid p(x) = 0\}$ and $Y_2 = \{x \in \overline{\mathcal{X}} \mid \exists \varepsilon > 0 : p(B_\varepsilon(x)) = 0\}$. Then we have

$$\int_{\mathcal{X} \setminus \mathcal{X}^+} p(x) \, dx = \int_{Y_1} p(x)dx + \int_{Y_2} p(x)dx = 0, \tag{32}$$

since, for each open $\varepsilon$-ball $B_\varepsilon(x)$, there exists a circle $Q(x) \subset B_\varepsilon(x)$ that contains $x$ and has rational radius and center coordinates. Thus, $Y_2$ can be covered by countably many such circles on which the integral of $p$ is zero. As a result, (32) implies that the elements of $\mathcal{X} \setminus \mathcal{X}^+$ do not affect the left-hand side of the first constraint and do not negatively impact the objective function or the second constraint. More precisely, for any feasible function $c : \mathcal{X} \to [0, 1]$, if

$$c'(x) = \begin{cases} c(x) & \text{if } x \in \mathcal{X}^+ \,, \\ 0 & \text{otherwise,} \end{cases} \tag{33}$$

then it holds $\int_{\mathcal{X}} R(x)c'(x)dx \leq \int_{\mathcal{X}} R(x)c(x)dx$, $\int_{\mathcal{X}} p(x)c'(x)dx = \int_{\mathcal{X}} p(x)c(x)dx$, and $\int_{\mathcal{X}} q(x)c'(x)dx \leq \int_{\mathcal{X}} q(x)c(x)dx$.

For the next considerations, we further distinguish three significant subsets of $\mathcal{X}^+$ with respect to a fixed optimal solution $c^* : \mathcal{X} \to [0, 1]$.

$$\mathcal{X}_0 = \left\{ x \in \mathcal{X}^+ \mid \exists \varepsilon > 0 : c^*(B_\varepsilon(x)) = 0 \right\} \,,$$
$$\mathcal{X}_1 = \left\{ x \in \mathcal{X}^+ \mid \exists \varepsilon > 0 : c^*(B_\varepsilon(x)) = 1 \right\} \,,$$
$$\mathcal{X}_2 = \left\{ x \in \mathcal{X}^+ \mid \forall \varepsilon > 0 : 0 < c^*(B_\varepsilon(x)) < 1 \right\} \,.$$

Note that

- $\mathcal{X}_0 \cup \mathcal{X}_1 \cup \mathcal{X}_2 = \mathcal{X}^+$,
- $c^*(B_\varepsilon(x)) = 0 \Rightarrow \forall 0 < \varepsilon' \leq \varepsilon : c^*(B_{\varepsilon'}(x)) = 0$, and

- $c^*(B_\varepsilon(x)) = 1 \Rightarrow \forall 0 < \varepsilon' \le \varepsilon : c^*(B_{\varepsilon'}(x)) = 1.$

To confirm the existence of suitable $\lambda$ and $\mu$, stated by the theorem for $c^*$, it is sufficient to demonstrate that the sets

$$A_0 = \{\varkappa(x) \mid x \in \mathcal{X}_0\}\,, \tag{34}$$
$$A_1 = \{\varkappa(x) \mid x \in \mathcal{X}_1\}\,, \tag{35}$$
$$A_2 = \{\varkappa(x) \mid x \in \mathcal{X}_2\} \tag{36}$$

are "almost" linearly separable. This necessitates finding a line $L \subset \mathbb{R}^2$ that includes $A_2$ and linearly separates $A_0 \setminus L$ and $A_1 \setminus L$. The existence of such an $L$ is ensured if

$$\dim \mathrm{span}\left((\mathrm{conv}(A_0) \cap \mathrm{conv}(A_1)) \cup A_2\right) < 2\,. \tag{37}$$

The proof continues by listing some general configurations of points from $A_0$, $A_1$, $A_2$ that contradict the optimality of $c^*$. These forbidden configurations will be used to confirm the validity of condition (37).

But we firstly establish estimates on $\int_{\varkappa^{-1}(B_\varepsilon(y))} q(x)c^*(x)dx$ and $\int_{\varkappa^{-1}(B_\varepsilon(y))} R(x)c^*(x)dx$, for each $y \in \mathcal{X}$ and $\varepsilon > 0$.

For a given $y \in \mathcal{X}$, let $\varkappa(y) = (a, b) = \left(\frac{q(y)}{p(y)}, \frac{R(y)}{p(y)}\right)$.

Since, for all $x \in \varkappa^{-1}(B_\varepsilon(y))$, it holds

$$a - \frac{\varepsilon}{2} \le \frac{q(x)}{p(x)} \le a + \frac{\varepsilon}{2}\,, \tag{38}$$

$$b - \frac{\varepsilon}{2} \le \frac{R(x)}{p(x)} \le b + \frac{\varepsilon}{2}\,, \tag{39}$$

we easily obtain

$$\left(a - \frac{\varepsilon}{2}\right)p(B_\varepsilon(y))c^*(B_\varepsilon(y)) \le \int_{\varkappa^{-1}(B_\varepsilon(y))} q(x)c^*(x)dx \le \left(a + \frac{\varepsilon}{2}\right)p(B_\varepsilon(y))c^*(B_\varepsilon(y))\,, \tag{40}$$

and

$$\left(b - \frac{\varepsilon}{2}\right)p(B_\varepsilon(y))c^*(B_\varepsilon(y)) \le \int_{\varkappa^{-1}(B_\varepsilon(y))} R(x)c^*(x)dx \le \left(b + \frac{\varepsilon}{2}\right)p(B_\varepsilon(y))c^*(B_\varepsilon(y))\,. \tag{41}$$

**Claim 1.1.** *Let $x_1, x_2 \in \mathcal{X}^+$, $\frac{q(x_1)}{p(x_1)} > \frac{q(x_2)}{p(x_2)}$, and $\frac{R(x_1)}{p(x_1)} > \frac{R(x_2)}{p(x_2)}$. If $c^*$ is optimal, then $x_1 \in \mathcal{X}_0$ or $x_2 \in \mathcal{X}_1$.*

*Proof of the claim.* We proceed by contradiction. Assume that $x_1 \in \mathcal{X}_1 \cup \mathcal{X}_2$ and $x_2 \in \mathcal{X}_0 \cup \mathcal{X}_2$. This implies the existence of an $\varepsilon > 0$ such that $c^*(B_\varepsilon(x_1)) > 0$ and $c^*(B_\varepsilon(x_2)) < 1$. Without loss of generality, choose $\varepsilon$ sufficiently small so that

$$\varepsilon \le \frac{q(x_1)}{p(x_1)} - \frac{q(x_2)}{p(x_2)} \tag{42}$$

and

$$\varepsilon \le \frac{R(x_1)}{p(x_1)} - \frac{R(x_2)}{p(x_2)}\,, \tag{43}$$

which further implies $B_\varepsilon(x_1) \cap B_\varepsilon(x_2) = \emptyset$.

For $i \in \{1, 2\}$, denote $B_i = B_\varepsilon(x_i)$, $a_i = \frac{q(x_i)}{p(x_i)}$, and $b_i = \frac{R(x_i)}{p(x_i)}$.

Define a function $c' : \mathcal{X} \to [0, 1]$ as follows:

$$c'(x) = \begin{cases} c^*(x) - \nu\frac{p(B_1)}{c^*(B_1)}c^*(x) & \text{if } x \in B_1, \\ c^*(x) + \nu\frac{p(B_1)}{p(B_2) - c^*(B_2)}(1 - c^*(x)) & \text{if } x \in B_2, \\ c^*(x) & \text{otherwise.} \end{cases}$$

Here, the parameter $\nu$ is defined as:

$$\nu = \min\left\{c^*(B_1), \frac{p(B_2)}{p(B_1)}(1 - c^*(B_2))\right\} > 0,$$

which ensures $0 < \nu\frac{p(B_1)}{c(B_1)} \le 1$ as well as $0 < \nu\frac{p(B_1)}{p(B_2)-c(B_2)} \le 1$, hence $c'$ is defined correctly.

By comparing $c^*$ and $c'$, we will show that $c^*$ is not optimal.

Observe that

$$\int_{\mathcal{X}} p(x)c'(x)dx - \int_{\mathcal{X}} p(x)c^*(x)dx =$$
$$- \nu\frac{p(B_1)}{c^*(B_1)}c^*(B_1) + \nu\frac{p(B_1)}{p(B_2)-c(B_2)}(p(B_2) - c^*(B_2)) = 0,$$

Moreover,

$$\int_{\mathcal{X}} q(x)c'(x)dx - \int_{\mathcal{X}} q(x)c^*(x)dx = -\nu\frac{p(B_1)}{c^*(B_1)}\int_{\varkappa^{-1}(B_1)} q(x)c^*(x)dx$$
$$+ \nu\frac{p(B_1)}{p(B_2)-c^*(B_2)}\int_{\varkappa^{-1}(B_2)} q(x)(1 - c^*(x))dx$$
$$\stackrel{(40)}{\le} -\nu\frac{p(B_1)}{c^*(B_1)}\left(a_1 - \frac{\varepsilon}{2}\right)c^*(B_1) + \nu\frac{p(B_1)}{p(B_2)-c^*(B_2)}\left(a_2 + \frac{\varepsilon}{2}\right)(p(B_2) - c^*(B_2))$$
$$= \nu p(B_1)(a_2 - a_1 + \varepsilon) \stackrel{(42)}{<} 0.$$

Analogously, we can derive

$$\int_{\mathcal{X}} R(x)c'(x)dx - \int_{\mathcal{X}} R(x)c^*(x)dx \le \nu p(B_1)(b_2 - b_1 + \varepsilon) \stackrel{(43)}{<} 0.$$

The relationships above contradict the optimality of $c^*$. ∎

**Claim 1.2.** *Let $x_1, x_2, x_3$ be distinct elements of $\mathcal{X}^+$, and let $\alpha_1, \alpha_2, \beta$ be non-negative real numbers such that $\alpha_1 + \alpha_2 = 1$, $\beta \ne 0$, and $\beta\varkappa(x_3) = \alpha_1\varkappa(x_1) + \alpha_2\varkappa(x_2)$. If $c^*$ is optimal, it holds:*

- *If $\beta < 1$, then either $x_3 \in \mathcal{X}_0$ or $\{x_1, x_2\} \cap \mathcal{X}_1 \ne \emptyset$,*

- *if $\beta > 1$, then either $x_3 \in \mathcal{X}_1$ or $\{x_1, x_2\} \cap \mathcal{X}_0 \ne \emptyset$.*

*Proof of the claim.* We will give a proof for $\beta < 1$ and note that the steps for $\beta > 1$ are analogous.

For $i \in \{1, 2, 3\}$, denote $\varkappa(x_i) = (a_i, b_i)$. Find an $\varepsilon > 0$ such that

$$\varepsilon < (1 - \beta)a_3, \tag{44}$$

$$\varepsilon < (1 - \beta)b_3, \tag{45}$$

and $B_\varepsilon(x_1), B_\varepsilon(x_2), B_\varepsilon(x_3)$ are pairwise disjoint. By contradiction, assume that $c^*(B_\varepsilon(x_1)) < 1$, $c^*(B_\varepsilon(x_2)) < 1$, and $c^*(B_\varepsilon(x_3)) > 0$.

To simplify the notation, for $i \in \{1, 2, 3\}$, let $B_i = B_\varepsilon(x_i)$, $c_i = c^*(B_i)$, and $p_i = p(B_i)$. Note that, according to the definition of $\mathcal{X}^+$, it holds $p_1, p_2, p_3 > 0$.

Define a selective function $c' : \mathcal{X} \to [0, 1]$ as follows:

$$c'(x) = \begin{cases} c^*(x) + \frac{\nu\alpha_1 p_3}{(1-c_1)p_1}(1 - c^*(x)) & \text{if } x \in B_1, \\ c^*(x) + \frac{\nu\alpha_2 p_3}{(1-c_2)p_2}(1 - c^*(x)) & \text{if } x \in B_2, \\ c^*(x) - \frac{\nu}{c_3}c^*(x) & \text{if } x \in B_3, \\ c^*(x) & \text{otherwise.} \end{cases}$$

Taking $\frac{z}{0} = \infty$ for any $z > 0$, the value of $\nu$ is given by:

$$\nu = \min\left\{c_3, \frac{p_1}{\alpha_1 p_3}(1 - c_1), \frac{p_2}{\alpha_2 p_3}(1 - c_2)\right\}.$$

It is easy to see that $\nu > 0$, and $0 < \frac{\nu\alpha_1 p_3}{(1-c_1)p_1}, \frac{\nu\alpha_2 p_3}{(1-c_2)p_2}, \frac{\nu}{c_3} \le 1$.

Now, observe that

$$\int_{\mathcal{X}} p(x)c'(x)dx - \int_{\mathcal{X}} p(x)c^*(x)dx = \int_{\varkappa^{-1}(B_1)} \frac{\nu\alpha_1 p_3}{(1-c_1)p_1}p(x)(1 - c^*(x))dx$$

$$+ \int_{\varkappa^{-1}(B_2)} \frac{\nu\alpha_2 p_3}{(1-c_2)p_2}p(x)(1 - c^*(x))dx - \int_{\varkappa^{-1}(B_3)} \frac{\nu p_3}{c_3}p(x)c^*(x)dx$$

$$= \nu\alpha_1 p_3 + \nu\alpha_2 p_3 - \nu p_3 = \nu p_3(\alpha_1 + \alpha_2 - 1) = 0.$$

Next, derive

$$\int_{\mathcal{X}} q(x)c'(x)dx - \int_{\mathcal{X}} q(x)c^*(x)dx = \int_{\varkappa^{-1}(B_1)} \frac{\nu\alpha_1 p_3}{(1-c_1)p_1}q(x)(1 - c^*(x))dx$$

$$+ \int_{\varkappa^{-1}(B_2)} \frac{\nu\alpha_2 p_3}{(1-c_2)p_2}q(x)(1 - c^*(x))dx - \int_{\varkappa^{-1}(B_3)} \frac{\nu p_3}{c_3}q(x)c^*(x)dx$$

$$\overset{(40)}{\le} \nu\alpha_1 p_3\left(a_1 + \frac{\varepsilon}{2}\right) + \nu\alpha_2 p_3\left(a_2 + \frac{\varepsilon}{2}\right) - \nu p_3\left(a_3 - \frac{\varepsilon}{2}\right)$$

$$= \nu p_3\left(\alpha_1 a_1 + \alpha_2 a_2 - a_3 + \frac{\varepsilon}{2}(\alpha_1 + \alpha_2) + \frac{\varepsilon}{2}\right)$$

$$= \nu p_3(\beta a_3 - a_3 + \varepsilon) \overset{(44)}{<} 0$$

Quite similarly,

$$\int_{\mathcal{X}} R(x)c'(x)dx - \int_{\mathcal{X}} R(x)c^*(x)dx \le \nu p_3(\beta b_3 - b_3 + \varepsilon) \overset{(45)}{<} 0.$$

$\blacksquare$

**Claim 1.3.** *Let $z_1, \ldots, z_n \in \mathcal{X}^+$ and $x \in \mathcal{X}^+$, and let $P$ be a convex polygon with vertices $\varkappa(y_1), \ldots, \varkappa(y_n)$, with $\varkappa(x)$ as its inner point. The function $c^*$ is not optimal if one of the following cases holds:*

1. *All $z_i$'s are from $\mathcal{X}_0$ and $x \in \mathcal{X}_1$.*

2. *All $z_i$'s are from $\mathcal{X}_1$ and $x \in \mathcal{X}_0$.*

*Proof of the claim.* For the first case, we identify an edge of the polygon $P$ that is first intersected by the half-line $H$ starting from the origin $(0, 0)$ and passing through the point $\varkappa(x)$ (see Figure 1a). This edge has vertices $\varkappa(y_1)$ and $\varkappa(y_2)$ for some $y_1, y_2 \in \{z_1, \ldots, z_n\}$. The points $\varkappa(x)$, $\varkappa(y_1)$, and $\varkappa(y_2)$ are non-collinear and satisfy $\beta\varkappa(x) = \alpha_1\varkappa(z_1) + \alpha_2\varkappa(z_2)$ for some non-negative real numbers $\alpha_1$, $\alpha_2$, and $\beta$, where $\alpha_1 + \alpha_2 = 1$ and $\beta < 1$. According to Claim 1.2, these points contradict the optimality of $c^*$.

Similarly, for the second case, we identify an edge of $P$ that is the last one to be intersected by the half-line $H$ (see Figure 1b). Let $\varkappa(y_1)$ and $\varkappa(y_2)$ be vertices of this edge for some $y_1, y_2 \in \{z_1, \ldots, z_n\}$. The points $\varkappa(x)$, $\varkappa(y_1)$, and $\varkappa(y_2)$ satisfy $\beta\varkappa(x) = \alpha_1\varkappa(y_1) + \alpha_2\varkappa(y_2)$ for some non-negative real numbers $\alpha_1$, $\alpha_2$, and $\beta$, where $\alpha_1 + \alpha_2 = 1$ and $\beta > 1$. Again, Claim 1.2 contradicts the optimality of $c^*$. $\blacksquare$

**Claim 1.4.** *If $x_1, x_2 \in \mathcal{X}_0$ and $x_3, x_4 \in \mathcal{X}_1$, and the line segments with endpoints $\varkappa(x_1)$, $\varkappa(x_2)$, and $\varkappa(x_3)$, $\varkappa(x_4)$ intersect at a point lying in the interior of both segments, then the function $c^*$ is not optimal.*

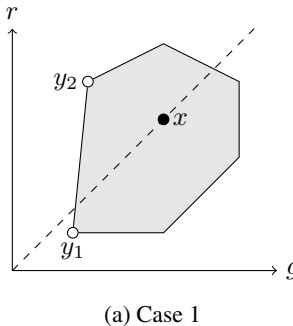

(a) Case 1

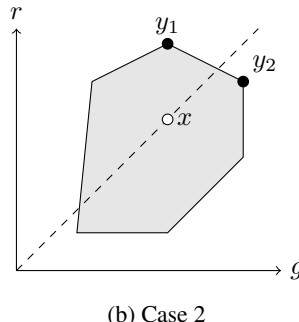

(b) Case 2

Figure 1: The infeasible configuration described by Claim 1.3. For simplicity, points are denoted by their pre-images from $\mathcal{X}^+$.

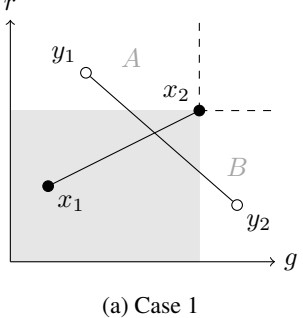

(a) Case 1

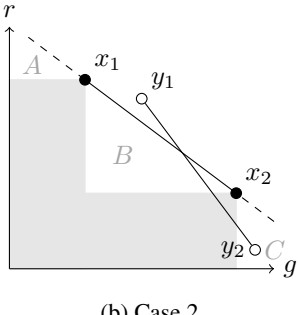

(b) Case 2

Figure 2: The infeasible configuration described by Claim 1.4.

*Proof of the claim.*

We consider two possible situations depicted in Figure 2.

First, we examine the scenario where one of the points $\varkappa(x_1)$, $\varkappa(x_2)$ is located lower-left relative to the other one, specifically, we assume $g(x_1) \leq g(x_2)$ and $r(x_1) \leq r(x_2)$, as illustrated in Figure 2a. According to Claim 1.1, the points $\varkappa(y_1)$, $\varkappa(y_2)$ cannot be situated within the shaded rectangular area. Since the line segments intersect, one of the points $\varkappa(y_1)$, $\varkappa(y_2)$ must be positioned upper-left from $\varkappa(x_2)$, while the other lies lower-right from $\varkappa(x_2)$ (these regions are labeled as $A$ and $B$, respectively, in Figure 1a). However, such an arrangement contradicts Claim 1.2.

Secondly, we consider the configuration depicted in Figure 2b. One of the points $\varkappa(y_1)$, $\varkappa(y_2)$ must lie within the half-plane bounded by the line passing through $\varkappa(x_1)$, $\varkappa(x_2)$, and containing the origin $(0, 0)$. Let's assume this point is $y_2$. As per Claim 1.1, $\varkappa(y_2)$ cannot reside in the shaded area. Within this considered half-plane, there are three remaining triangular regions denoted as $A$, $B$, and $C$. Nevertheless, Claim 1.2 prohibits $\varkappa(y_2)$ from lying within region $B$ (by applying the claim to the triple $x_1$, $x_2$, and $y_2$), region $C$ (consider the triple $y_1$, $y_2$, and $x_2$), as well as region $A$ (consider the triple $y_1$, $y_2$, and $x_1$). ∎

**Claim 1.5.** *If $z_1, z_2 \in \mathcal{X}_2$ and $\varkappa(z_1)$, $\varkappa(z_2)$ are distinct points, then the slope of the line $L$ passing through $\varkappa(z_1)$ and $\varkappa(z_2)$ is not positive. Moreover, the open half-plane determined by $L$ and the origin $(0, 0)$ does not contain any point from $A_1 \cup A_2$, and similarly, the opposite open half-plane does not contain any point from $A_0 \cup A_2$.*

*Proof of the claim.* If the slope of $L$ is positive, as shown in Figure 3a, then the points $\varkappa(z_1)$ and $\varkappa(z_2)$ conform to Claim 1.1, implying that $c^*$ is not optimal.

If the slope of $L$ is not positive, referring to Figure 3b, we deduce that the open half-plane $H_0$ does not contain any point $\varkappa(x)$ from $A_0 \cup A_2$. By Claim 1.1, such a point $P$ cannot be in the shaded area. It neither can be in areas $A$, $B$, or $C$ because in each case, the points $x$, $z_2$, and $z_1$, or $z_1$, $z_2$, and $x$, or $z_1$, $x$, and $z_2$ would conform to Claim 1.2.

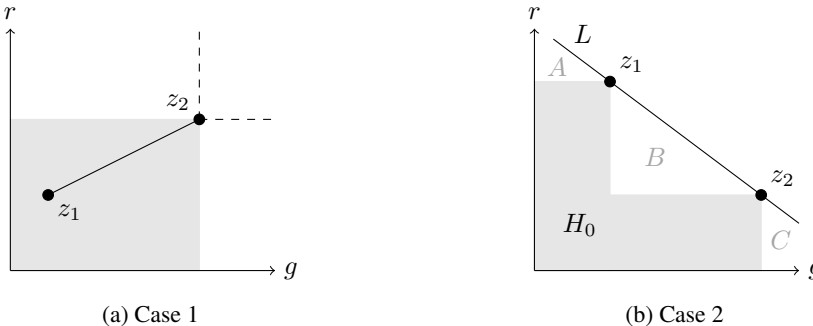

(a) Case 1            (b) Case 2

Figure 3: The infeasible configuration described by Claim 1.5.

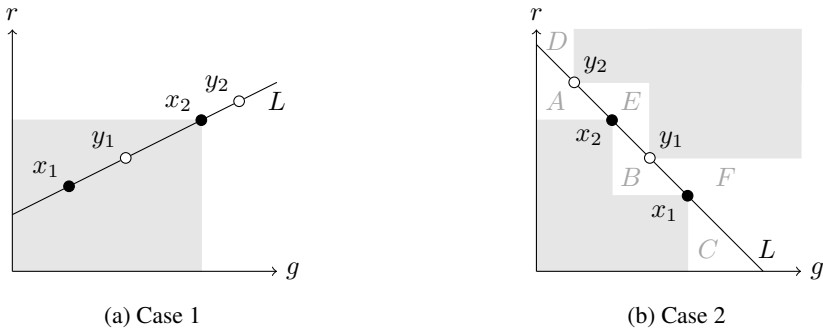

(a) Case 1            (b) Case 2

Figure 4: The infeasible configuration described by Claim 1.6.

Similarly, it can be argued that a point from $A_1 \cup A_2$ cannot lie in the opposite open half-plane.

∎

**Claim 1.6.** *Let $x_1, x_2 \in \mathcal{X}_1$, $y_1, y_2 \in \mathcal{X}_0$, $z \in \mathcal{X}_2$. If $\varkappa(x_1)$, $\varkappa(x_2)$, $\varkappa(y_1)$, and $\varkappa(y_2)$ are distinct points lying on a line $L$ in such a way that $\varkappa(y_1)$ is between $\varkappa(x_1)$ and $\varkappa(x_2)$, and $\varkappa(x_2)$ is between $\varkappa(y_1)$ and $\varkappa(y_2)$, then $\varkappa(z)$ lies on $L$ unless the function $c^*$ is not optimal.*

*Proof of the claim.*

Firstly, observe that the configuration in Figure 4a is not admissible for an optimal $c^*$, because $\varkappa(y_1)$ is located in the shaded area containing points to the left and down from $\varkappa(x_2)$ (Claim 1.1 applies here).

Hence, it suffices to inspect the case with the opposite slope of $L$ depicted in Figure 4b. Note that, without loss of generality, we assume that the highest point on $L$ is $\varkappa(y_2)$. The case when the highest point is either $\varkappa(x_1)$ or $\varkappa(x_2)$ is handled similarly. If $c^*$ is optimal, then $\varkappa(z)$ cannot lie in the two shaded areas (Claim 1.1). The remaining area is split into 6 different parts by $L$, denoted from $A$ to $F$. If $\varkappa(z)$ lies in any of these parts, then we can always select two elements from $\{x_1, x_2, y_1, y_2\}$ that together with $z$ conform to Claim 1.2. These selections are as follows:

$$A\colon z, y_1, x_2, \qquad B\colon x_1, x_2, z, \qquad C\colon z, y_1, x_1,$$
$$D\colon z, x_2, y_2, \qquad E\colon y_1, y_2, z, \qquad F\colon z, x_2, y_1.$$

∎

We are ready to confirm condition (37), by listing all the potential infeasible cases and showing that they contradict the optimality of $c^*$.

1. $\dim \mathrm{span}(\mathrm{conv}(A_0) \cap \mathrm{conv}(A_1)) = 2$. We distinguish three subcases.

   Assume that, for some $x \in \mathcal{X}_0$, $\varkappa(x)$ is in the interior of $\mathrm{conv}(A_0) \cap \mathrm{conv}(A_1)$. By Lemma 2, there is a convex polygon $P$ with vertices in $\varkappa(\mathcal{X}_1)$ such that the interior of $P$ contains $x$. Apply Claim 1.3.

Next, assume that $x$ belongs to $\mathcal{X}_1$ instead of $\mathcal{X}_0$. Analogously, find a convex polygon $P'$ with vertices in $\varkappa(\mathcal{X}_0)$, containing $x$ in its interior. Apply Claim 1.3.

Finally, if $\operatorname{int}(\operatorname{conv}(A_0) \cap \operatorname{conv}(A_1)) \cap (A_0 \cup A_1) = \emptyset$, take an arbitrary $x \in \operatorname{int}(\operatorname{conv}(A_0) \cap \operatorname{conv}(A_1))$. By Carathéodory's theorem, there exists a simplex $S_0$ (a line segment or triangle) with vertices in $A_0$ such that $\varkappa(x) \in \operatorname{int}(S_0)$. Similarly, there exists a simplex $S_1$ (a line segment or triangle) with vertices in $A_1$ such that $\varkappa(x) \in \operatorname{int}(S_1)$. It is not difficult to see that the boundary line segments of $S_0$ and $S_1$ intersect, meaning there are two intersecting line segments, one with endpoints in $A_0$, and the other with endpoints in $A_1$. Hence, Claim 1.4 contradicts the optimality of $c^*$.

2. $\dim \operatorname{span}(A_2) = 2$. There exist $x_1, x_2, x_3 \in \mathcal{X}_2$ such that $\varkappa(x_1), \varkappa(x_2), \varkappa(x_3)$ are non-collinear. Such a configuration is prohibited by Claim 1.5.

3. $\dim \operatorname{span}(\operatorname{conv}(A_0) \cap \operatorname{conv}(A_1)) = 1$, $\dim \operatorname{span}(A_2) \le 1$ and $\dim \operatorname{span}((\operatorname{conv}(A_0) \cap \operatorname{conv}(A_1)) \cup A_2) = 2$. There are $x_1, x_2 \in \mathcal{X}_0$, $x_3, x_4 \in \mathcal{X}_1$, $x_5 \in \mathcal{X}_2$ conforming to Claim 1.6.

4. $\dim \operatorname{span}(\operatorname{conv}(A_0) \cap \operatorname{conv}(A_1)) = 0$, $\dim \operatorname{span}(A_2) = 1$, and $\dim \operatorname{span}((\operatorname{conv}(A_0) \cap \operatorname{conv}(A_1)) \cup A_2) = 2$. There are $z_1, z_2 \in \mathcal{X}_2$ such that $\varkappa(z_1)$ and $\varkappa(z_2)$ conform to Claim 1.5, i.e., the line $L$ passing through $\varkappa(z_1)$ and $\varkappa(z_2)$ determines two opposite half-planes, $H_0$ and $H_1$. By Claim 1.5, $A_0 \subseteq H_0$ and $A_1 \subseteq H_1$. This means that $\operatorname{conv}(A_0) \cap \operatorname{conv}(A_1) \subseteq L$, which contradicts the assumption $\dim \operatorname{span}((\operatorname{conv}(A_0) \cap \operatorname{conv}(A_1)) \cup A_2) = 2$.

$\square$

