# OpenReview forum: "Constrained Binary Decision Making"
_NeurIPS.cc/2024/Conference — NeurIPS 2024 poster_

### Official Review · Reviewer_CrjX · 2024-07-11

**Soundness:** 3
**Presentation:** 3
**Contribution:** 3
**Rating:** 6
**Confidence:** 2

**Summary:**

The authors proposed an optimization problem which has specific form of solutions that can be leveraged to solve various types of binary statistical decision making problems.

**Strengths:**

The formulation of the optimization is general and there are many applications in binary statistical decision making problem.

**Weaknesses:**

The motivation for characterizing the optimal solution is quote the authors: "This exmaple underscores the advantages of understanding the structure of the optimal solution to underlying BDM problems". So it seems reasonable to me to add some experiments to validate the claim.

**Questions:**

As above.

**Limitations:**

adequately addressed

---

> ### Author Rebuttal · Authors · 2024-08-04
>
> R: *The motivation for characterizing the optimal solution is quote the authors: "This example underscores the advantages of understanding the structure of the optimal solution to underlying BDM problems". So it seems reasonable to me to add some experiments to validate the claim.*
>
> A: Recent papers [7] and [16] independently provided empirical evidence that the optimal strategy for SCOD outperforms the heuristic strategy SIRC [23].

---

> > ### Comment · Area_Chair_NCJq · 2024-08-12
> > **Reviewer response?**
> >
> > Reviewer CrjX, could you please review the authors' response and see whether it addresses your questions? Please acknowledge having done so in a comment. Thanks.

---

> > ### Comment · Reviewer_CrjX · 2024-08-12
> > **Reply to rebuttal**
> >
> > Sorry for the late reply. I thank the reviewer for their clarification. Their reply addresses my concern, and I would like to keep my score.

---

### Official Review · Reviewer_oCzV · 2024-07-12

**Soundness:** 3
**Presentation:** 4
**Contribution:** 4
**Rating:** 8
**Confidence:** 3

**Summary:**

The paper titled "Constrained Binary Decision Making" presents a comprehensive framework for binary statistical decision making (BDM), a critical area in both classical statistics and modern machine learning. The authors formulated BDM problems as constrained optimization tasks and provide a detailed characterization of the optimal solutions. The paper covers well-known and recent BDM problems, demonstrating the applicability of their generic approach to derive optimal decision strategies.

**Strengths:**

- This paper is very well-written paper, and easy to follow.

- The authors proposed an interesting framework that encompasses several popular BDM problems and presented solid proof.

**Weaknesses:**

- The authors claimed "Conversely, skipping the optimal strategy derivation and using heuristic rules, such as the SIRC strategy from the original SCOD paper [23], can lead to sub-optimal performance". I think it would be interesting (but not required) to conduct experiments to compare the proposed approach against the one in [23].

**Questions:**

- In Lemma 1, I do not know what $\pi(x)$ and $\rho(x)$ stand for. Should they be $p(x)$ and $q(x)$, respectively?

**Limitations:**

The authors claimed that "The paper presents a theoretical result on an optimization problem, hence, in its essence, it has no limitations." I think in the **Extensions** section, I'm wondering that whether the proposed approach can be adapted to handle the case where more than one **$\geq$** constraints exist, could this be a limitation of the current framework?

---

> ### Author Rebuttal · Authors · 2024-08-04
>
> R: *The authors claimed "Conversely, skipping the optimal strategy derivation and using heuristic rules, such as the SIRC strategy from the original SCOD paper [23], can lead to sub-optimal performance". I think it would be interesting (but not required) to conduct experiments to compare the proposed approach against the one in [23].*
>
> A: Recent papers [7] and [16] independently provided empirical evidence that the optimal strategy for SCOD outperforms the heuristic strategy SIRC [23].
>
> R: *The authors claimed that "The paper presents a theoretical result on an optimization problem, hence, in its essence, it has no limitations." I think in the Extensions section, I'm wondering that whether the proposed approach can be adapted to handle the case where more than one constraints exist, could this be a limitation of the current framework?*
>
> A: We agree with the comment. Since the proposed extension is an unproven hypothesis, we will list it among the paper's limitations.
>
> R: *In Lemma 1, I do not know what $\pi(x)$ and $\rho(x)$ stand for. Should they be $p(x)$ and $q(x)$, respectively?*
>
> A: Yes, it is a typo, these functions should be $p(x)$ and $q(x)$.

---

> > ### Comment · Reviewer_oCzV · 2024-08-09
> >
> > Thanks for addressing my comments. I will keep my score unchanged.

---

### Official Review · Reviewer_bcuc · 2024-07-23

**Soundness:** 3
**Presentation:** 3
**Contribution:** 3
**Rating:** 6
**Confidence:** 4

**Summary:**

This paper studies a class of optimality criteria for what it calls "binary decision making" which is basically binary classification but with a randomized classifier. It characterizes the solutions for thes criteria, recovering some known results and also establishing new ones. The paper is entirely theoretical.

**Strengths:**

The main strength of the paper is that it provides a very general treatment of optimality criteria for binary classification , where the criteria include an objective to be minimized, and also one or two constraints to be satisfied. It subsumes prior work on statistical optimality in "selective classification." The proof appears to be original, nontrivial, and, for the most part, clearly written.

**Weaknesses:**

The main weakness concerns the impact. Of course impact on practitioners is desired, but even the theoretical impact was unclear to me at times. I think this stems from how the paper is motivated. The motivation uses the notion of "selective classification," where you have a classifier and a selection function, and your optimality criterion concerns both of them. But the actual results in the paper only discuss the selection function, so there is a disconnect. So if the authors could clarify that issue, and also make a sincere effort to explain realistic settings where the novel criteria are potentially useful, that would go a long way toward resolving the main weakness.

The other notable weakness is that the motivation involves a classifier h and a selection function c, but then h goes away at some point.

Below I include a section by section list of comments that I made while reading the paper.

Introduction: Writing could be improved. For example, the Neyman-Pearson lemma is introduced in both the first and second paragraphs. Selective classification is not so well known, and it would be helpful to define it here, and explain why it is an important class of criteria. Most importantly, BDM is never precisely defined.

*** What is the impact of the contribution? Does it help us train selective classifiers? Some optimal solutions seem to require knowledge of the underlying distributions. In line 165 it is stated that the results are “potentially useful for specific applications”, but no concrete examples are given. Thus the impact could be more convincingly argued. This is probably the main weakness.

The sentence “Therefore, learning a detector from examples involves effectively approximating the likelihood ratio and then tuning the decision threshold” is seemingly false, as you just need the decision boundary

Reference [17]: “Person” should be “Pearson”

** Sec 2: What is the learning problem? Do we just get data from the “in distribution”?

The statement “We assume the classifier h was designed to minimize the prediction loss … ” is unclear – usually one chooses h to minimize an expected loss. But it is not clear what that expectation (joint distribution) is at this point in the presentation.

Eqn. (7): The notation p(x,y), to me, assumes x and y are jointly discrete. Better to just use E for expectation and eliminate the integral and sum.

Eqn. (10): missing “)”

*** The review of BDM problems discussed the optimal c, but the optimal h is not mentioned.

Selective risk: notation varies, e.g., R^S vs R_S

** The final example assumes that pi is known or can be estimated. Well, it will not be known in practice, and if estimated, the estimate won’t equal the true pi. Thus, it would be important to understand how having an estimate of pi impacts the usefulness of this case.

There are several papers that cover classes of optimal prediction functions in different setups, and I think it would be appropriate to mention this work and discussion connections. For example,
Consistent Binary Classification with Generalized Performance Metrics, Oluwasanmi O. Koyejo, Nagarajan Natarajan, Pradeep K. Ravikumar, Inderjit S. Dhillon
Harikrishna Narasimhan, Rohit Vaish, and Shivani Agarwal. On the statistical consistency of plugin classifiers for non-decomposable performance measures.
Krzysztof Dembczy´nski, Wojciech Kotłowski, Oluwasanmi Koyejo, and Nagarajan Natarajan. Consistency analysis for binary classification revisited.
Clayton Scott, A Generalized Neyman-Pearson Criterion for Optimal Domain Adaptation

One common theme is that the optimal decision rule is always a likelihood ratio test. If it’s not, it means the criterion being optimized is not sensible. So you could use that to frame your contribution.

I would also recommend looking a the simple idea presented in the section on “Birdsall’s Insight” in Statistical Signal Processing by Louis Scharf.

Another potentially related paper is Classification with confidence by Jing Lei

Sec 3

In the setup, the functions are assumed to be Lebesgue measurable. This means \mathcal{X} should be a Euclidean space, which has not been indicated.

In the statement of Lemma 1, the notations rho(x) and pi(x) appear out of nowhere.

*** The function R depends on c, whereas before it depended on c and h. An explanation is needed. The whole motivation involved h, but now h has been lost.

Sec 4

Line 235: “useful variants” -> “special cases”

Line 235: “whose solutions can be derived” – the full solution would require specification of tau(x) and lambda, so instead you could say that you are characterizing the form of the solution.

Problem 1: mention that this does not fully recover the NP lemma because in the NP lemma, tau is a constant.

In the discussion of problems 4 and 5, I *think* you mean to say that these can be transformed to (25), not (24).

** Can you give an example of where it does not suffice for tau(x) to be a constant? In all of the examples, tau(x) is neglected.

Proofs

Line 337: what notion of dimension is meant? Do you mean it has positive Lebesgue measure on R^2?

Line 348: why is C’ a compact set?

Line 353: Need to discuss why pi is measurable, and presumably here A is a Borel set.

Line 357-358: It’s clear that such x do not impact the first constraint, but it’s not clear that it doesn’t impact the objective function or second constraint.

Line 360: contains pi(x)

**Questions:**

In the previous box, a ** or *** next to a comment means I would be interested in hearing the authors' response, with *** indicating higher priority. Responses to other comments are also welcome.

**Limitations:**

There are no anticipated negative societal impacts.

---

> ### Author Rebuttal · Authors · 2024-08-04
>
> **Priority questions**
>
> R: *What is the impact of the contribution? Does it help us train selective classifiers? Some optimal solutions seem to require knowledge of the underlying distributions. In line 165 it is stated that the results are “potentially useful for specific applications”, but no concrete examples are given. Thus the impact could be more convincingly argued. This is probably the main weakness.*
>
> A: Our paper characterizes optimal strategies for various BDM decision problems, such as those used to define optimal selective classifiers. Although the formulations and derived optimal strategies rely on known data distributions, our results have two immediate practical impacts at least. First, learning involves finding a decision strategy within a predefined hypothesis space based on data. Knowing the optimal strategy's form allows to define a hypothesis space that includes the optimal strategy, essential for a statistically consistent algorithm. For reference, see the introduction, where we discuss prior SCOD works that used heuristically chosen hypothesis spaces that did not contain the optimal strategy, and a follow-up paper that improved results by modifying the heuristic to match the optimal rule. Second, our results allow the construction of plug-in optimal strategies for various problems.
>
> R: *Sec 2: What is the learning problem? Do we just get data from the “in distribution”?*
>
> A: Section 2 does not address any learning problems. Its goal is to provide examples of BDM problems and their optimal solutions (strategies). These formulations and strategies assume the data distribution is known. The significance of knowing the optimal strategy's form for designing learning algorithms is discussed in the previous answer.
>
> R: *The review of BDM problems discussed the optimal c, but the optimal h is not mentioned.*
>
> A: Our paper focuses on finding the optimal $c$ by solving various instances of the BDM problem. Except for the Neyman-Pearson problem, the example problems also involve finding the predictor $h$. However, in all the examples determining the optimal strategy for $h$ is straightforward. In all case, Bayes predictor $h^*(x)= {\rm argmin_{y'}} \sum_{y}p(y\mid x)\ell(y,y')$ is optimal due to additivity of the risk $R^S$, which allows $h$ to be optimized independently for each instance $x$.
>
> R: *The final example assumes that pi is known or can be estimated. Well, it will not be known in practice, and if estimated, the estimate won’t equal the true pi. Thus, it would be important to understand how having an estimate of pi impacts the usefulness of this case.*
>
> A: Yes, the formulation in Sec 2.6 is applicable only if $\pi$ is known or can be estimated. If $\pi$ is unknown, the formulations in Sec 2.4 and 2.5 should be used. While analyzing the sensitivity of the solution to the OOD ratio $\pi$ in Sec 2.6 could be an interesting topic for future research, it is not the focus of our current paper.
>
> R: *The function R depends on c, whereas before it depended on c and h. An explanation is needed. The whole motivation involved h, but now h has been lost.*
>
> A: Thanks for point this out this inconsistency. We will include $h$ and emphasize that finding the optimal $h$ is not an issues (see the answer above).
>
> R: *Can you give an example of where it does not suffice for tau(x) to be a constant? In all of the examples, tau(x) is neglected.*
>
> A: There is a degenerate case when the score $s(x)$ equals the threshold $\lambda$ for all $x\in {\cal X}$, and $\tau(x)$ acts as a selection function. In this scenario, the points $(R(x)/p(x),q(x)/p(x))$ lie on a line $L$. Consequently, $\tau(x)$ equals 1 for a line segment subset of $L$, and 0 otherwise. In practice, the degenerate cases occur when the instance space ${\cal X}$ is finite.
>
> R: *In the statement of Lemma 1, the notations rho(x) and pi(x) appear out of nowhere.*
>
> A: Yes, it is a typo, these functions should be $p(x)$ and $q(x)$.
>
> **Proofs**
>
> R: *Line 337: what notion of dimension is meant? Do you mean it has positive Lebesgue measure on $\mathbb{R}^2$?*
>
> A: It is the dimension of $\rm{span}(A)$ in the vector space $\mathbb{R}^2$.
>
> R: *Line 348: why is $C'$ a compact set?*
>
> A: We will provide details to show that $C'$ is both complete and totally bounded, i.e. compact. Essentially, a Cauchy sequence of feasible $c$'s cannot converge to a function that violates any of the constraints by some $\varepsilon > 0$.
>
> R: *Line 353: Need to discuss why $\pi$ is measurable, and presumably here $A$ is a Borel set.*
>
> A: Agreed, a discussion is required. We need to define $p(A)$, $c(A)$ only for those $A$ that are epsilon-balls in $\mathbb{R}^2$. Then, we need to show that $\pi^{-1}(A)$ is a measurable subset of ${\cal X}$. It is fulfilled since this subset is determined by mesuarable functions derived from the functions $p(x), q(x), R(x)$.
>
> R: *Line 357-358: It’s clear that such $x$ do not impact the first constraint, but it’s not clear that it doesn’t impact the objective function or second constraint.*
>
> A: For such $x$ we can set $c^*(x)=0$ since this does not impact the first constrain and does not worsen the criterion or the second constraint. We will give a more detailed explanation.
>
> R: *Line 360: contains $\pi(x)$*
>
> A: You are right.
> ____
>
> Thank you for highlighting these issues. We will revise and enhance the proof accordingly.
>
> We will also apply the other suggested corrections to the main text.

---

> > ### Comment · Reviewer_bcuc · 2024-08-08
> >
> > I would like to thank the authors for responding to my review. I will maintain my score of weak accept. I have no doubt the authors can improve the paper, but there are enough changes that I would want to re-review the entire paper before making a stronger recommendation.

---

### Official Review · Reviewer_aUyF · 2024-07-30

**Soundness:** 3
**Presentation:** 3
**Contribution:** 3
**Rating:** 6
**Confidence:** 3

**Summary:**

This paper characterizes optimal solutions to a class of binary statistical decision-making (BDM) problems. The problem recovers as special cases the likelihood-ratio problem and variants of classification with rejection problems. The optimal solutions characterized in this paper coincide with the known solutions to the aforementioned problems, and thanks to this paper, new solutions are found to some new variants of the BDM problem.

**Strengths:**

I like the paper. It is written very cleanly. The motivation is clear. I also enjoy how the paper starts with a slow pace, gives a "tutorial" on BDMs, and then uses this to connect with their results -- since the Neyman-Pearson lemma is very old and there exist many variants of the notation, this way of presentation was particularly useful. I like Theorem 1 and Lemma 1, and the proofs use smart techniques (e.g., even Lemma 1 looks simple, but the construction of c* from an optimal c is very clear in my view).

**Weaknesses:**

In my view, the following points are the weaknesses of the paper and I kindly invite the authors to address them:
- In general, problem (17) is an infinite-dimensional linear program. There is a huge literature, many books, etc., on this. These resources devote significant time to characterizing optimal solutions and the existence of feasibility. Could the authors add more discussion on why the existing literature on infinite linear programs (ILPs) cannot be used here? One can look at "Linear programming in infinite-dimensional spaces: theory and applications" by Nash and Anderson, or related papers.
- Section 2.5: It is said optimal solutions to the given problem are unknown. But this problem variant is also defined in this paper. Can the authors also motivate "it is not known, and can *not* be derived by the existing machinery"? Currently, the flow is ad-hoc in the sense that the authors define a new problem, discuss there is no known solution, and derive one that resembles the solutions to similar problems from the earlier subsections.
- Section 2.5: In line 145, it is said SCOD is using different units in the objective function. However, there are many multi-objective problems with different units. For example, portfolio optimization involves balancing between returns and covariances, which have different units. This problem needs further motivation in my view.
- Please make the notation consistent: (14) is using $R^\mathrm{S}(h,c)$ but earlier it was $R^\mathrm{S}(c)$. I think the former is better, but please be consistent across the paper. Furthermore, Lemma 1 has $\rho$ and $\pi$, which I believe should be $q$ and $p$.
- After Theorem 1, before immediately presenting Lemma 1, can the authors discuss Theorem 1? I like the idea of having a single function to use in the optimal solution characterization via piece-wise functions (as this resembles existing results), however, equations (20) and (21) do not give much intuition currently.
- Some of the computational challenges are presented as it is easy. For example line 215: "Once these ratios are known, determining the score $s$ involves finding the multiplier $\mu$". Can the authors discuss how to find $\mu$? Similarly "one **only** needs to find the scalars $\mu$ and $\lambda$". Same for line 224, "we obtain a similar problem but with only one constraint.": how do you solve this problem, there is no mention.
- The paper would greatly benefit from some numerical experiments. Currently, it is highly theoretical and there are strong assumptions like the knowledge of underlying distributions. There are mentions of cross-validation (or estimation from held-out data), which is more of a computational task than statistical in my view, and I would like to see some mini-experiments on this.

Some minor typos and issues:
- Line 70: "is" generated from the in-distribution
- Line 91: "to distinguishes" has a typo
- Line 107, "reflects the probability of accepting an input sample": I don't think this is about an input sample. Isn't $\phi$ returning the measure of the set that we "reject"? The input of $\phi$ is $c$, not $x$.
- Equation (10): Missing comma ","
- Some references to (15) should be (14) instead. Examples are line 149, and 172.
- Line 157: "solvers" should be "solves"? That said "the strategy" should be "a strategy" if there can be alternative optima.

**Questions:**

- At the beginning of the paper, it is said that the BDM problem involves Lebesgue measurable functions. It is not clear to me whether all the literature is specifically focused on basically density functions. Can the authors confirm this and add a more in-depth discussion, please?
- Why is there a focus on "robust" keyword in this work? Especially because we assume data-generating distributions like $p_I$ are known, there is no robustness. I am a little confused. Some of the work on the "classification with rejection" literature has motivation for robustness, but I don't see how this is generalized for BDMs in general.
- Are defining the distributions as $p: \mathcal{X} \mapsto \mathbb{R}_+$ the formal way? Is this how the literature defines them?
- Line 77, "only when prediction uncertainty is minimal": I don't understand this sentence. Also, there are uses of $p(x, y)$ and $p(y \mid x)$. Please formally define them.
- Can you discuss the measurability of the functions, such as terms in (7)?
- Equation (12): Aren't there work in the literature that also constrains $\phi(c)$?
- Line 232 says "the combinatorial complexity of the proof" will be increased if we have more constraints. How do the authors know the proof will still go through? If the authors are convinced, I would at least propose a high-level proof sketch in the appendix.

**Limitations:**

The limitations are addressed to some extent, and the checklist is complete. However, further focus on the computational issues about the proposed model should be highlighted.

---

> ### Author Rebuttal · Authors · 2024-08-04
>
> **Weaknesses**
>
> R: *In general, problem (17) is an infinite-dimensional linear program (ILP). There is a huge literature on this.*
>
> A: We agree that BDM is an instance of ILP, enabling the use of tools like duality and KKT conditions. However, it is unclear if these tools can provide the same characterization of the optimal decision strategy for BDM or simplify our proof. We do not see an immediate, straightforward application. Furthermore, existing proofs related to the Neyman-Pearson problem, to our knowledge, do not consider ILP. We will include a discussion on this topic.
>
> R: *Section 2.5: It is said optimal solutions to the given problem are unknown. But this problem variant is also defined in this paper. Can the authors also motivate "it is not known, and can not be derived by the existing machinery"?*
>
> A: Our goal is to demonstrate the generic nature of the proposed framework. Therefore, we introduced novel modifications to the existing SCOD problem (12). These modifications, which include additional constraints and an optimized function in the denominator when defining precision, are more complex but can be readily resolved. We will clarify this in the text.
>
> R: *Section 2.5: It is said SCOD is using different units in the objective function. However, there are many multi-objective problems with different units. This problem needs further motivation.*
>
> A: We do not intend to criticize multi-objective problems. Instead, we aim to demonstrate that alternative formulations, which may be more appropriate in certain cases, exist and can be readily solved.
>
> R: *Please make the notation consistent in (14), Lemma 1.*
>
> A: Thanks for pointing this. We will make the notation consistent. There is a typo, these functions should be $p(x)$ and $q(x)$.
>
> R: *Can the authors discuss Theorem 1?*
>
> A: We will add a discussion on equations (18)-(21). The sets ${\cal X}^{<}$, ${\cal X}^{>}$ are subsets of ${\cal X}$ separated by the score function; they ignore insignificant $x\in {\cal X}$ for which $p(x)=0$.
>
> R: *Some of the computational challenges are presented as it is easy. Can the authors discuss how to find $\mu$ and $\lambda$"?*
>
> A: The method for finding unknown multipliers varies by problem. In some cases, such as the optimal strategy for the SCOD problem (13), the multiplier can be calculated from the problem's input parameters. If no explicit formula exists, the standard approach is to tune the parameters using calibration data. Notably, tuning one or two parameters, while challenging, is significantly easier than tuning an unknown function, which would be required without the optimal strategy characterization.
>
> **Questions**
>
> R: *At the beginning of the paper, it is said that the BDM problem involves Lebesgue measurable functions. It is not clear to me whether all the literature is specifically focused on basically density functions. Can the authors confirm this and add a more in-depth discussion, please?*
>
> A: At least in all the example problems from Sec 2, the functions involved are either probability density functions (p.d.f.s) or p.d.f.s multiplied by loss functions, all of which are Lebesgue measurable.
>
> R: *Why is there a focus on "robust" keyword?*
>
> A: We use the term "robust" to indicate that the tools proposed in the paper i) require very weak assumptions for application and ii) are applicable to a broad range of BDM problems.
>
> R: *Are defining the distributions as $p\colon{\cal X}\rightarrow\mathbb{R}_+$ the formal way? Is this how the literature defines them?*
>
> A: In defining the functions used in the problem formulations, we specify only their domains and co-domains, such as $p\colon{\cal X}\rightarrow\mathbb{R}_+$. Probability density functions (p.d.f.s) must also be Lebesgue integrable with an integral equal to one. We assume this well-known standard definition, therefore, we have not included it.
>
> R: *Line 77, "only when prediction uncertainty is minimal": I don't understand this sentence. Also, there are uses of $p(x,y)$ and $p(y|y)$. Please formally define them.*
>
> A: We agree that the sentence is confusing. A clearer explanation would be: "We will explore six different BDM instances to develop a selector $c$ that admits an input sample $x$ for classification with $h$ only when the prediction uncertainty is below an acceptable threshold."
>
> We use $p(x,y)$ to denote joint probability distribution of random variables $X$ and $Y$. In our setup, these variables be either discrete or continuous; however, in the example problems, $Y$ is always discrete. We use $p(y \mid x)$ to represent the conditional probability of a discrete random variable $Y$ given another random variable $X$.
>
> R: *Can you discuss the measurability of the functions, such as terms in (7)?*
>
> A: Yes, this is a valid point. We should state that functions such as $c$ and $\ell$ are assumed to be measurable.
>
> R: *Equation (12): Aren't there work in the literature that also constrains $\phi(c)$?*
>
> A: The constraint in (12) imposes a lower bound on the true positive rate, ${\rm tpr}(c)$, which is defined identically to the coverage $\phi(c)$ . In the context of SCOD, this is referred to as the "true positive rate," whereas in selective classification, it is called "coverage."
>
> R: *Line 232 says "the combinatorial complexity of the proof" will be increased if we have more constraints. How do the authors know the proof will still go through? If the authors are convinced, I would at least propose a high-level proof sketch in the appendix.*
>
> A: To be cautious, we will present the extension as a hypothesis for future exploration. The Appendix will include a remark suggesting potential avenues for generalization.
>
> R: *The limitations are addressed to some extent, and the checklist is complete. However, further focus on the computational issues about the proposed model should be highlighted.*
>
> A: We will extend the discussion by explaining that tuning the multipliers needs be addressed for each problem separately.

---

> > ### Comment · Reviewer_aUyF · 2024-08-07
> > **Thank you & I am concerned**
> >
> > I would like to thank the authors for their rebuttal.
> >
> > Although I appreciate the authors going through each of my comments, I would like to kindly state that this rebuttal is not satisfactory to me. None of my comments are addressed in any depth. To give some examples:
> >
> > - For multi-objective discussion, the authors say "We do not intend to criticize multi-objective problems" but regardless of what the authors intend, it's just an informal and unusual motivation to revise multi-objective programs just because of differences in units.
> >
> > - More importantly, to my understanding, the known proofs of Neyman-Pearson lemma do not use any infinite-dimensional LP field. But this paper does take that approach. It is still ok if such a result is known out there, but I don't think "We do not see an immediate, straightforward application." is a convincing argument. I am not even sure KKT conditions are common in ILP literature. In general, I was hoping to see more than "We will include a discussion on this topic."
> >
> > Finally, I would like to strongly recommend not using statements like "mu and lambda tuning is significantly easier". It can still be intractable in many settings. Just because we have less variables to tune should not imply this paper presents a tractable approach.

---

> > > ### Author Response · Authors · 2024-08-09
> > > **Detailed answers.**
> > >
> > > R: Although I appreciate the authors going through each of my comments, I would like to kindly state that this rebuttal is not satisfactory to me. None of my comments are addressed in any depth.
> > >
> > > A: Since there was no priority list, we addressed all the reviewer's questions. However, due to the 6,000-character limit, we had to keep our responses concise.
> > >
> > > R: For multi-objective discussion, the authors ...
> > >
> > > A: The inability to assign the same physical units to individual decisions (and hidden states) often prevents straightforward weighting of multiple errors with different meanings. As a result, constraints on specific errors are commonly used, especially when certain errors need explicit control in specific applications. This approach is standard in multi-criteria decision-making and not unique to the SCOD problem. For example, similar strategies are applied in selective classification, as seen in Sections 2.2 and 2.3, which offer alternatives to the traditional cost-based reject-option classifier [3]. We applied this same strategy to the SCOD problem.
> > >
> > > However, please note that our paper's primary focus is not on revising existing SCOD formulations. We use them to demonstrate the versatility and usefulness of our framework.
> > >
> > > R: More importantly, to my understanding, the known ...
> > >
> > > A: We initially considered using the principle of Lagrange duality to address our ILP problem, as it offers a way to establish optimality conditions that could potentially characterize the solutions. We applied duality to finite domains ${\cal X}$, transforming the ILP into an LP, which gave us some insight into the general solution form. However, extending this approach to an arbitrary domain ${\cal X}$ and functions $R$, $p$, $q$ (with finite integrals) would require a more general duality theorem, and even then, deducing the desired result would be far from straightforward due to the increased complexity of the optimality conditions. Given these challenges and inspired by techniques used in related work (e.g., [9, 16, 17] and others on the Neyman-Pearson problem), we decided to pursue a direct proof instead. This approach provided us with solid insights, leaving only the technical details to be completed. While we do not dismiss the possibility that ILP techniques could yield the same result, potentially with a simpler proof, we believe that this path is not as straightforward as it might seem.
> > >
> > > R: Finally, I would like to strongly recommend not using ...
> > >
> > > A: We respectfully disagree with the suggestion that optimizing two or three scalars is not "significantly easier" than optimizing an entire function. In all existing examples, tuning these parameters by discretizing their values and performing an exhaustive search, though not necessarily optimal, has proven effective in practice.

---

### Official Review · Reviewer_ALNx · 2024-08-10

**Soundness:** 3
**Presentation:** 3
**Contribution:** 3
**Rating:** 6
**Confidence:** 3

**Summary:**

The paper presents a framework for binary statistical decision-making (BDM), where decisions are made between two states based on statistical evidence. The authors introduce a constrained optimization problem formulation for BDM, involving integrals of Lebesgue measurable functions, and provide a detailed characterization of the optimal decision strategies. The paper encompasses a wide range of existing and newly proposed BDM problems as specific instances of the presented framework, demonstrating how to derive optimal strategies for these problems. This framework aims to simplify the process of solving both established and novel BDM problems, which are fundamental to many machine learning algorithms.

**Strengths:**

Originality:
The paper introduces a novel and general framework that unifies various binary decision-making problems under a single constrained optimization formulation. This generalization is a significant contribution, as it provides a robust mathematical tool that can be applied to a wide range of existing and newly proposed BDM problems.
Quality:

The paper is rigorous in its theoretical treatment, providing detailed proofs and a clear characterization of the optimal strategies for BDM problems. The use of Lebesgue measurable functions and the generality of the framework allow it to cover both discrete and continuous instances without requiring differentiability of decision and loss functions.

Clarity:
I like it a lot that the authors provided lots of examples in section 2. The paper is well-organized, with clear explanations of the problem formulations and the derivation of optimal strategies. The inclusion of examples from classical statistics and machine learning applications enhances the clarity and relevance of the work. The theoretical results are presented in a step-by-step manner, making the paper accessible to readers with a background in optimization and decision theory.

Significance:
The significance of the paper lies in its potential to impact a broad range of applications in machine learning and statistical decision-making. By providing a unified framework for BDM problems, the paper simplifies the process of deriving optimal strategies, which can be crucial for designing efficient algorithms in areas such as selective classification, out-of-distribution detection, and hypothesis testing.

**Weaknesses:**

Limited Empirical Validation:
The paper primarily focuses on the theoretical aspects of BDM and does not provide empirical validation of the proposed framework. While the theoretical results are strong, it would be beneficial to see experimental evaluations that demonstrate the practical effectiveness of the framework in real-world scenarios.


Computational Complexity:
The proposed optimization problems involve integrals of Lebesgue measurable functions, which may be computationally expensive to solve, especially for high-dimensional data. The paper does not fully address the computational challenges associated with solving these problems, which could be a barrier to practical implementation.

**Questions:**

Have the authors considered conducting empirical experiments to validate the proposed framework? Demonstrating the practical effectiveness of the optimal strategies in real-world BDM problems could significantly strengthen the paper's contributions.
Handling Unknown Distributions:

How would the framework handle cases where the underlying data distributions are unknown or difficult to estimate? Are there potential extensions or modifications that could make the framework more robust to distributional uncertainty?
Computational Efficiency:

---

> ### Author Response · Authors · 2024-08-14
> **Authors answers to reviewer's questions**
>
> R: Have the authors considered conducting empirical experiments to validate the proposed framework? ...
>
> A: Our paper focuses solely on providing a mathematical tool for deriving optimal strategies in various BDM problems. Our theorem implies these optimal strategies cannot perform worse than any alternatives, and in the worst case, the improvement might be small. However, there are instances where BDM problems initially addressed with heuristic strategies were later shown to perform significantly better when optimal strategies were identified and empirically validated.
>
> A recent example from the machine learning field is the BDM in the SCOD problem (please, see paragraph 3 of the introduction). Initially solved with the SIRC selection strategy [23], it was later shown to be suboptimal in [16], with empirical evidence supporting this. Similarly, the Neyman-Pearson problem in statistics, one of the earliest BDM examples, offers ample empirical proof that the likelihood ratio outperforms alternative methods for separating two distributions.
>
> R: How would the framework handle cases where the underlying data distributions are unknown or difficult to estimate? Are there potential extensions or modifications that could make the framework more robust to distributional uncertainty?
>
> A: Yes, distribution uncertainty can be addressed by formulating the BDM problem. The SCOD problem is a good example of this. By definition, there is no clear sample of Out-Of-Distribution (OOD) data. However, the optimal strategy shows that modeling the ratio of OOD to In-Distribution (ID) samples is sufficient for making optimal decisions (see Equation 13). Methods for estimating this OOD/ID ratio, such as using an unlabeled mixture of ID and OOD data, have been discussed in [16] and other papers.

---

### Author Rebuttal · Authors · 2024-08-04

We thank all reviewers for their efforts and valuable comments. Our responses are submitted separately for each review.

---

### Decision · Program_Chairs · 2024-09-25

**Decision:**

Accept (poster)

**Comment:**

The authors develop a rigorous, theoretical approach to binary classification with a randomized classifier. The reviewers find the approach new and the proof techniques interesting. The expert reviewers do have a long list of changes that they're requesting and I would appreciate if the authors would please pay careful attention to these changes before the camera-ready deadline.